# Targeted Therapies for the Evolving Molecular Landscape of Acute Myeloid Leukemia

**DOI:** 10.3390/cancers13184646

**Published:** 2021-09-16

**Authors:** Khashayar Ahmadmehrabi, Ali R. Haque, Ahmed Aleem, Elizabeth A. Griffiths, Gregory W. Roloff

**Affiliations:** 1Department of Medicine, Loyola University Medical Center, Maywood, IL 60153, USA; khashayar.ahmadmehrabi@luhs.org (K.A.); Ali.Haque@luhs.org (A.R.H.); ahmed.aleem@luhs.org (A.A.); 2Department of Medicine, Roswell Park Comprehensive Cancer Center, Buffalo, NY 14203, USA; Elizabeth.Griffiths@roswellpark.org

**Keywords:** acute myeloid leukemia, therapeutics, targeted therapies, relapse, transplantation

## Abstract

**Simple Summary:**

Acute myeloid leukemia (AML) is a predominately fatal blood cancer. For a period of forty years, treatment options for AML remained relatively stagnant. Recently, multiple new agents have been approved. In this review, we discuss considerations surrounding the use of these newly approved therapies. We outline the molecular profiles of AML disease status and highlight subsets of patients for whom therapies are best suited based on available data.

**Abstract:**

Despite considerable growth in our understanding of the heterogeneous biology and pathogenesis of acute myeloid leukemia (AML) in recent decades, for nearly forty years, little progress was gained in the realm of novel therapeutics. Since 2017, however, nine agents have been FDA-approved for patients with AML in both the upfront and relapsed/refractory (R/R) settings. Most of these compounds function as inhibitors of key cell cycle enzymatic pathways or mediators of leukemic proliferation and survival. They have been approved both as single agents and in combination with conventional or reduced-intensity conventional chemotherapeutics. In this article, we review the molecular landscape of de novo vs. R/R AML and highlight the potential translational impact of defined molecular disease subsets. We also highlight several recent agents that have entered the therapeutic armamentarium and where they fit in the AML treatment landscape, with a focus on *FLT3* inhibitors, *IDH1* and *IDH2* inhibitors, and venetoclax. Finally, we close with a survey of two promising novel agents under investigation that are poised to enter the mainstream clinical arena in the near future.

## 1. Introduction

In 2021, actuarial assessment by the Surveillance, Epidemiology and End Results (SEER) Program of the United States National Cancer Institute estimates acute myeloid leukemia (AML) to be the most common acute leukemia in adults, with an anticipated 20,240 new diagnoses and 11,400 deaths [1]. AML is a disease of hematopoietic progenitor cells wherein the acquisition of genetic mutations and/or chromosomal rearrangements help drive the expansion of immature myeloid populations [2,3]. Additionally, inherited predisposition syndromes driven by germline risk variants have recently gained increased recognition by oncologists [4]. Patients most commonly present with symptoms related to bone marrow failure which cause anemia, increased bruising and bleeding (from thrombocytopenia) and enhanced susceptibility to infections (from numerical or functional neutropenia). Diagnosis is made through evaluation of the patients’ clinical history, careful physical examination, routine laboratory assessment, and bone marrow aspiration and biopsy to distinguish AML, characterized by a blast percentage of more than 20%, from other diseases of bone marrow failure such as myelodysplasia or aplastic anemia [5].

After an AML diagnosis is rendered, prognostication into favorable, intermediate, or adverse risk groups is made based on the presence of defined cytogenetic and molecular aberrations to help predict response to induction chemotherapy and risk for relapse (European Leukemia Net (ELN) classification) [6]. Recently, the advent of next-generation sequencing (NGS) has allowed for comprehensive profiling of the AML mutational landscape at the individual patient level. Testing at the time of initial diagnosis can be done in a simple binary fashion (i.e., an assay for the presence or absence of a clinically relevant mutation such as *FLT3* ITD) or using broader NGS panels to identify mutational networks providing increased precision within an individual ELN-defined risk category [7]. One example of a more complex molecular category includes the identification of co-mutations in tyrosine kinase/chromatin modifier genes (*ASXL1*/*2*, *EZH2*, *KDM6A*, *BCOR*, and *BCORL1*) and/or cohesion genes (*RAD21*, *STAG2*, *SMC1A*, and *SMC3*) which represent an adverse combination with a high (~50%) incidence of relapse within the previously categorized good risk ELN category of patients harboring the core binding factor rearrangement t(8;21) [8,9].

The picture becomes even more complicated when analyzing differential cytogenetic and molecular landscapes in patients with R/R AML. Multiple analyses at the chromosomal, mutational, and immunophenotypic levels have depicted the complexity of disease resistance and progression [10,11,12]. The biological networks that underlie these processes, while incompletely characterized, will undoubtedly represent exciting avenues for therapeutic development in years to come. In this review, we overview recently approved and emerging targeted therapies for the evolving molecular landscape of de novo and R/R acute myeloid leukemia.

## 2. First: The Problem with “Fitness”

Several novel AML drugs approved in the past 24 months (Figure 1) (venetoclax, glasdegib, ivosidenib monotherapy, oral azacitidine) are intended for use in those unfit to undergo intensive induction therapy or proceed to stem cell transplantation (SCT) once remission is achieved. These agents expand the treatment selection options for older patients. This is significant because retrospective analyses have revealed that about half of patients >65 years of age with a new diagnosis of AML receive no active therapy [13,14]. Venetoclax in combination with hypomethylating agents (HMA) such as azacitidine or decitabine, venetoclax in combination with low dose cytarabine, and glasdegib in combination with low dose cytarabine have each won FDA approval on the grounds of the benefit they impart in patients in whom intensive induction therapy is precluded or who simply present for diagnosis above the age of 75 years [15,16].

Unfortunately, “fitness” is highly subjective without a uniform definition or harmonized collection of clinical or laboratory thresholds below which patients are deemed “unfit”. While Eastern Cooperative Oncology Group (ECOG) and Karnofsky Performance Status (PS) scoring systems are frequently employed, the decision to label a patient fit or unfit for induction therapy is left to physicians, who possess inherent biases [17,18]. Palmieri and colleagues have recently provided a comprehensive overview of how clinicians might navigate various prediction/scoring systems to augment clinical decision making based on AML patient fitness in the era of new drugs [19].

Unfortunately, “fitness” is highly subjective without a uniform definition or harmonized collection of clinical or laboratory thresholds below which patients are deemed “unfit”. While Eastern Cooperative Oncology Group (ECOG) and Karnofsky Performance Status (PS) scoring systems are frequently employed, the decision to label a patient fit or unfit for induction therapy is left to physicians, who possess inherent biases [17,18]. Palmieri and colleagues have recently provided a comprehensive overview of how clinicians might navigate various prediction/scoring systems to augment clinical decision making based on AML patient fitness in the era of new drugs [19].

Unfortunately, “fitness” is highly subjective without a uniform definition or harmonized collection of clinical or laboratory thresholds below which patients are deemed “unfit”. While Eastern Cooperative Oncology Group (ECOG) and Karnofsky Performance Status (PS) scoring systems are frequently employed, the decision to label a patient fit or unfit for induction therapy is left to physicians, who possess inherent biases [17,18]. Palmieri and colleagues have recently provided a comprehensive overview of how clinicians might navigate various prediction/scoring systems to augment clinical decision making based on AML patient fitness in the era of new drugs [19].

Unfortunately, “fitness” is highly subjective without a uniform definition or harmonized collection of clinical or laboratory thresholds below which patients are deemed “unfit”. While Eastern Cooperative Oncology Group (ECOG) and Karnofsky Performance Status (PS) scoring systems are frequently employed, the decision to label a patient fit or unfit for induction therapy is left to physicians, who possess inherent biases [17,18]. Palmieri and colleagues have recently provided a comprehensive overview of how clinicians might navigate various prediction/scoring systems to augment clinical decision making based on AML patient fitness in the era of new drugs [19].

Honest appraisal of patient inclusion demographics in the two flagship venetoclax trials (discussed at length below) would reveal that 84% of patients who received venetoclax + HMA had a ECOG 0–1 and 0% of patients had an ECOG of 3–4 [20]. Amongst those evaluated in the study combining venetoclax and low dose cytarabine, 71% were ECOG 0–1 [21]. The act of identifying individuals unfit for conventional induction therapy rests largely on the assumption that unacceptable levels of treatment-related mortality (TRM) might be observed in these patients if traditional therapeutic avenues were pursued. TRM, defined as death attributable to adverse events of the therapy in patients by day 28, is based on multiple variables, including age, but also measures of organ function such as albumin, platelet count, and creatinine. Using a TRM calculator, Estey, Karp, and colleagues demonstrated that a hypothetical 75 year old patient with reduced renal function (creatinine 1.5 mg/dL), who might theoretically be labeled unfit and treated with reduced intensity options, would in fact be expected to have <10% TRM with traditional induction measures [22]. A recently published 11-year, multi-site analysis of 1292 patients split into retrospective (from 2008–2012) and prospective (from 2013–2017) cohorts found that less-intensive therapies offered no benefits in terms of overall survival, quality of life, or functional status when compared with traditional intensive regimens. An initial signal demonstrating improved overall survival in the traditional intensive therapy group failed to remain statistically significant when accounting for physician perception of patient performance status, indicating the need for future randomized study [23].

As further delineated by Cook and colleagues, the consequences of labeling those who could potentially receive more aggressive therapies as unfit and subsequently including them in single-arm clinical studies are multidimensional. Most acutely, it prohibits investigators from being able to measure the outcomes in patients who might have been fit for intense regimens all along. Inclusion of potentially fit patients into such trials also creates a blind spot when trying to ascertain the tolerability of novel agents intended for unfit populations. Namely, such patient selection bias could reasonably be expected to inflate the presumed efficacy of such novel therapies, since these relatively healthier patients would likely tolerate higher doses of the drug under investigation than a truly “unfit” population of unselected patients presenting with AML. Lastly, setting a standard for approval based on nonrandomized, single-arm studies allows drug companies to forego execution of more rigorous trial designs in which the agent in question is tested against traditional regimens. If the results of such trials demonstrated that a new agent was superior, they would indeed be transformative and might obviate the need for such subjective measures of fitness [24].

## 3. Molecular Landscape of De Novo AML vs. R/R AML

The development and integration of high-throughput genomics tools, namely, NGS technologies, has helped define molecular disease subclasses and track clonal composition over time. Seminal work by Papaemmanuil and colleagues has provided a blueprint depicting genomic subgroups of AML by describing the clustering of canonical driver mutations across traditionally partitioned cytogenetic and ELN risk categories [25]. Depending on whether AML development presents de novo or represents evolution from an antecedent hematologic disorder, the constellation of somatic mutational events has been observed to follow some general patterns. Most notably, mutations in so-called “DTA genes” (*DNMT3A*, *ASXL1*, *TET2*) are now widely recognized contributors to age related clonal hematopoiesis and observed in as many as 10% of individuals over age 60, the majority of whom will never be afflicted with myeloid malignancy [26,27]. Follow-up studies analyzing banked blood samples of AML patients years before diagnosis have more deeply characterized the mutational composition that seems to be important for AML development. Namely, historic samples from those who went on to develop AML show that these patients have greater numbers of mutations, higher variant allele frequencies, and distinct patterns of mutations in specific genes, including *TP53*, *IDH1*, and *IDH2* [28,29].

While most patients with AML able to be induced with conventional chemotherapy attain full morphological remission, a substantial proportion of patients experience leukemic relapse. The genetic drivers underpinning leukemic relapse are characterized by competition among subclones within complex mutational hierarchies. A major focus of studies on the biology of AML relapse has been to discriminate whether leukemic subclones responsible for relapse and death contribute to the diagnostic tumor burden or evolve via classic Darwinian models due to direct mutagenic effects of chemotherapy on specific cell populations [30,31]. Patient-derived xenografts have arisen as a powerful model to study these phenomena due to the retention of leukemic stem cell capacity after transplantation into mice [32]. Sandén and colleagues have carried out serial transplantation assays to characterize subclone biology over time. Patient AML samples, having undergone cytogenetic and mutational profiling, were engrafted into immunodeficient mice and allowed to mature over a period of 15 months before longitudinal NGS was performed. Several distinct patterns of clonal evolution were observed. A majority of disease engraftments demonstrated significant shift in the clonal composition over time, with only 26% of cases retaining the initial diagnostic architecture. About half (48%) of cases demonstrated clonal expansion from a minor subclone that had variant allele frequencies of ~5% at the time of patient diagnosis. Other rare clones that emerged at patient relapse were either undetectable or detectable at the level of background (“noise”) in clinical samples and only revealed by transplantation xenograft models [33]. The detection of underappreciated diagnostic clones suggests a higher level of disease heterogeneity at initial presentation than previously believed and indicates that genetic variability from diagnosis to relapse also relies on intraclonal competitive evolution within individuals.

Most AML patients with intermediate or high-risk disease who achieve a first complete remission and who are not encumbered by significant medical comorbidities are referred for SCT in an effort to maximize long term survival. Unfortunately, post-transplant relapse in AML remains common and is associated with particularly inferior outcomes [34]. Early investigation using single-nucleotide polymorphism arrays or conventional cytogenetic techniques has associated post-transplant AML relapse with the acquisition of chromosome duplications, deletions, or loss of heterozygosity in the relapse sample compared with the diagnostic counterpart [35,36]. More recently, with the growing integration of NGS-based disease profiling at academic medical centers, the mutational patterns that emerge at relapse have been described with increasing molecular detail. These studies have also favored models that describe relapse as spawning from subclonal populations present at diagnosis as opposed to the emergence of novel clones with mutational compositions that were undetected in the diagnostic specimens [37] (Figure 2, top panel).

By utilizing optimized exome sequencing, recent seminal work by Christopher [31] and colleagues has disentangled one of the mechanisms by which AML cells escape the beneficial graft-versus-leukemia effect imparted by allograft. Since it has long been postulated that patients who relapse after transplant do so, in part, by evading immune surveillance mechanisms, the investigators carried out DNA and RNA sequencing on paired samples from AML patients at time points before and after post-SCT relapse. In all, >200 genes known play critical roles in immune function were differentially expressed across the pre-and-post transplant settings. The authors found that half (17/34) of the patients with post-SCT relapse demonstrated downregulation of MHC class II genes (Figure 2, middle panel). Notably, the cellular effects of MHC-II downregulation were overcome by treatment with gamma-interferon. Because expression of MHC II molecules was not diminished in pretransplant samples, it is likely these changes may have been mediated by epigenetic mechanisms and further studies to explore potential re-sensitization of relapsed cells to graft-versus-tumor immune surveillance are warranted.

In addition to NGS-based mutational profiling, flow cytometric approaches have also helped to mold our understanding of molecular evolution of AML relapse vs. diagnosis. Ho and colleagues have demonstrated through serial dilution and transplantation assays the evolution of AML stem cell properties that predominate at relapse vs. initial diagnosis. By collecting patient samples at presentation and after relapse and carrying out limited dilution assays into immunodeficient mice after flow cytometric cell sorting, they found a 9- to 90-fold increase in AML stem cell frequency at relapse when compared with diagnosis. Results were most profound for a leukemia stem cell enriched population that was CD123^+^CD34^+^CD38^−^ [10] (Figure 2, bottom panel).

## 4. Focus on Inhibiting Single-Gene Mutations

### 4.1. FLT3 Inhibition

FMS-like tyrosine kinase 3 (*FLT3*) is a tyrosine kinase receptor found on the outer membrane of healthy myeloid cell populations. In AML, expression of the receptor exceeds that of normal myeloid progenitors and is also known to harbor distinctive mutations in the juxtamembrane domain and a downstream activating kinase moiety [38]. When bound by its ligand, the *FLT3* receptor dimerizes to allow phosphorylation of the internal domain to generate a cascade of intracellular signals driving cell proliferation, growth and division. Pathogenic *FLT3* mutations allow for constitutive activation and autophosphorylation of the tyrosine kinase domain in a ligand-independent manner. Compared with cytogenetically normal AML, *FLT3* mutant leukemias are characterized by a higher incidence of disease recurrence after conventional cytoreductive therapy, shorter time to relapse, and lower overall survival [38].

Two well-defined activating mutations hold clinical relevance: the *FLT3* Internal Tandem Duplicate (ITD) and *FLT3* tyrosine kinase domain (TKD) point mutation. *FLT3* ITD mutations are among the most common mutations in AML, comprising about 25% of cases, and point mutations in the *FLT3* TKD (D835 or I836) make up 5–10% of cases [39]. While the exact prognostic significance of *FLT3* TKD mutant AML remains unclear, because this class of disease as a whole portends poorer outcomes, the development of potent tyrosine kinase inhibitors quickly emerged as an area of medical necessity.

Preclinical studies of midostaurin, also known as PCK412, uncovered the drug’s potential therapeutic effects by demonstrating its antiproliferative activity against various tumors [40]. Midostaurin and its metabolites induce cell cycle arrest via potent inhibitory effects on a number of different kinases, most notably FLT3, c-KIT, PDGFR, and protein kinase C. Initial testing in humans demonstrated reduction in size of several solid tumors in addition diminution of circulating lymphocytes and monocytes while remaining generally well tolerated via oral route of administration [41].

These early data inspired formalized testing of midostaurin in the AML R/R setting. A phase II trial published by Stone and colleagues in 2005 demonstrated that 70% of participating patients experienced a 50% or greater reduction in bone marrow or peripheral blast counts [42]. The natural extension of these results was to next test the addition of *FLT3* inhibitors to upfront chemotherapy for patients with de novo *FLT3* mutated AML. A phase 1b study by Stone et al., assessed various dosing and scheduling formulations of midostaurin combined with cytarabine and daunorubicin induction chemotherapy and cytarabine post-remission therapy in adults aged 18–60 with new onset disease [43].

The landmark appraisal of midostaurin’s efficacy came with the phase III CALGB RATIFY trial [44]. Patients were evenly randomized to midostaurin or placebo on days 8–21 of upfront therapy with a traditional cytarabine and anthracycline backbone. Up to two cycles of induction and up to four cycles of high-dose cytarabine consolidation in combination with midostaurin was allowed, followed by maintenance midostaurin treatment for up to twelve 28-day cycles. Of note, patients were permitted to pursue allogeneic transplant and would discontinue midostaurin therapy at this time. Midostaurin decreased risk of death by 22% when compared with placebo and was observed for all *FLT3* subtypes. Rates of complete remission were marginally increased in the midostaurin group compared with placebo (58.9% for those who received midostaurin vs. 53.5% for those who received placebo) despite significant improvement in overall survival. This might be explained in part by the fact that more patients in the midostaurin group proceeded to SCT after initial treatment success.

Although now FDA approved alongside use of a companion PCR assay for *FLT3*, several questions surrounding the use of midostaurin remain. The most obvious pertains to the age of patients assessed in RATIFY, those aged 18–60. While the drug was FDA approved with a broad indication to be used for FLT3 mutant disease, AML presents at an average age 68 and therefore the effects of adding midostauin to standard induction therapy in a population burdened by additional comorbidities has not been well defined. As pointed out by Lai and colleagues, a large ongoing trial (NCT03512197) aimed at assessing the off-target kinase effects of midostaurin in patients without FLT3 mutation was designed without an age limit and is expected to provide insights on the tolerability of midostaurin in older AML patients [45].

Gilteritinib is a potent selective inhibitor of *FLT3* autophosphorylation. Through its specificity for FLT3, gilteritinib has shown high rates of antileukemic activity as a monotherapy in R/R *FLT3* mutated AML patients while maintaining a satisfactory safety profile [46]. In an open-label, randomized large phase III study known as the ADMIRAL trial, patients with R/R FLT3 mutant AML were randomized in a 2:1 manner to receive gilteritinib monotherapy or salvage chemotherapy. Randomization was guided by the intensity of upfront therapy patients received before relapse and subsequent response. These regimens included a high intensity treatment group (mitoxantrone/etoposide/cytarabine and FLAG-IDA) and a low intensity group (low-dose cytarabine and azacitidine) [47].

The primary endpoints measured were overall survival and the percentage of patients who had complete remission with full or partial hematologic recovery. Notable secondarily endpoints included event free survival and the percentage of patients that were able to achieve complete remission. A total of 37.1% of the gilteritinib-treated patients were alive at one year, as compared with 16.7% of the patients in the chemotherapy arm. This translated to a median overall survival of 9.3 months in the gilteritinib-treated patients as compared with 5.6 months in the chemotherapy arm (*p* < 0.001). The hazard ratio for death was found to be 0.64 (95% confidence interval [CI], 0.49 to 0.83) in the gilteritinib treatment group. Furthermore, complete remission with full or partial hematologic response was noted in 34% of the gilteritinib-treated patients as opposed to 15.3% in the chemotherapy arm, translating to a risk difference of 18.6%. Complete remission was noted in 21% of the gilteritinib-treated patients, as opposed to 10.5% in the chemotherapy arm [47]. The significant improvements noted in overall survival and the percent of patients able to achieve full remission led to the FDA approval of gilteritinib as monotherapy in the R/R setting, along with use of a companion diagnostic PCR assay [48]. A multitude of follow up studies are currently ongoing examining the use of gilteritinib as maintenance therapy after attainment of first remission (NCT02927262), as maintenance therapy following allogeneic transplant (NCT02997202), or in combination with azacitadine for upfront therapy in patients unable to receive standard induction (NCT02752035).

As is the case with other single agent targeted therapies, patients treated with gilteritinib in the ADMIRAL went on to experience relapse after an initial response to therapy. Follow up laboratory studies by McMahon and colleagues have revealed mechanisms of disease escape under the selective pressures of FLT3 inhibition [49]. By targeted NGS conducted on patient samples at baseline and at the time of progression while on gilteritinib treatment, activating mutations in *RAS/MAPK* pathway genes emerged as a major mechanism enabling drug resistance and disease progression. Additionally, observed in this cohort were secondary *FLT3* F691L “gatekeeper” mutations. Detailed single-cell sequencing analysis further revealed heterogeneous trajectories of clonal evolution and resistance patterns in the cohort studied, which included the acquisition of RAS mutations in *FLT3* mutant populations, the expansion of *FLT3* mutant-negative clusters, or both occurring in synchrony.

Recently published molecular follow-up of the RATIFY study also shines light on mechanisms of FLT3 inhibition under the selective of midostaurin. Of note, 59% of patients in the midostaurin arm on RATIFY achieved complete remission, and half of them went on to experience disease relapse. By performing targeted *FLT3* analysis and whole exome sequencing on 54 paired patient samples at diagnosis and relapse, Schmalbrock and colleagues have described high rates (46%) of relapse with FLT3-ITD-negative clones [50]. In these patients, proliferative advantage also seemed to be driven by acquired mutations in common signaling pathways (MAPK). Additionally remarkable was the observation of mutational stability of the initial *FLT3 ITD* variant in 32% of relapsed patients, suggesting more complex networks of disease escape bypassing FLT3 signaling.

The frontiers of FLT3 inhibition continue to be investigated beyond the FDA approval of midostaurin and gilteritinib. Notably, multiple trials are ongoing evaluating the safety and efficacy of FLT3 inhibitors in AML affecting pediatric and young adult patients [51]. The majority of work to date has focused on sorafenib, including Children’s Oncology Group AAML1031 (NCT01371981) and Children’s Oncology Group ADVL0413 (NCT01445080). These studies represent a response to an urgently unmet clinical need, as pediatric patients harboring FLT3-mutant disease have generally fared poorly, with 30% event-free survival noted across cooperative group trials [52]. If approved in this population, the addition of FLT3 inhibitors will add to recent approvals for of gentuzumab ozogamicin and liposomal cytarabine and daunorubicin for newly diagnosed and secondary-AML, respectively.

### 4.2. IDH1/2 Inhibition

Acquired mutations in genes isocitrate dehydrogenase 1 (*IDH1*) and *IDH2* are found in nearly 20% of AML cases [53]. Three major, recurrent mutations in the *IDH* genes at *IDH1*-R132, *IDH2*-R172, and *IDH2*-R140 are associated with the formation of *R*-2-hydroxyglutarate by promoting the inactivation of α-ketoglutarate-dependent enzymes [54]. Once formed, *R*-2-hydroxyglutarate acts as an oncometabolite and propagates the disabling of DNA methylation and cellular differentiation. *IDH*-mutated cells preferentially produce the (R)-enantiomer of 2-HG, and accumulation of *R*-2-HG propogates leukemogenesis by causing differentiation arrest [55]. This differentiation block can be overcome by restoration of *R*-2-HG to normal cellular levels [56]. The prognostic significance of these mutations on molecular risk stratifications is the topic of debate in the field and was not included in the Updated 2017 ELN guidelines, but we feel that mutations in *IDH1/2* confer intermediate risk disease [57].

Enasidenib, a targeted *IDH2* inhibitor, was the first drug in its class to gain FDA approval based on results from a 2017 phase I/II trial in patients with *IDH2* mutant R/R AML wherein about 19.6% of patients achieved a complete remission with a median duration of response of 5.6 months. While most responses with enasidenib are not durable and overall survival was 8.8 months, it might reasonably be employed to extend remission as a bridge to transplant, since about half of patients who achieved a CR successfully proceeded to allograft [57,58]. Additional trials evaluating the insertion of enasidenib into the therapeutic armamentarium are ongoing, with two studies evaluating enasidenib as post-transplant maintenance therapy (NCT03515512 and NCT03728335) and another evaluating the combination of enasidenib with azacitadine in newly diagnosed *IDH2* mutant disease (NCT02677922). At a cost of over USD 29,000 per month of therapy, it will be prudent to remain aware of whether enasidenib, especially in the maintenance setting (where therapy duration is often indefinite in those not proceeding to transplant), proves superior to other, more cost-effective options.

Recently work has also demonstrated the promise of *IDH2* inhibition in patients with myelodysplastic syndromes. By analyzing a subgroup of the landmark phase I/II trial, Stein and colleagues reported responses in 9 of 17 patients (53%) bearing *IDH2* mutation. The average duration of response was 9.2 months and median overall survival 16.9 months. Of note, several patients who derived benefit from enasidenib had previously received 2+ lines of therapy, including hypomethylating agents, thus raising potential for formal investigation in a space with limited options (NCT03383575) (NCT03744390) [59].

Ivosidenib, a selective IDH1 inhibitor, gained entry into the AML treatment toolbox after the publication of results from phase 1 study in which 125 R/R AML patients received 500 mg of ivosidenib once daily. Rates of CR plus CR with partial hematologic response were observed to be 30.4% with a mediation duration of response of 8.2 months [60]. Observed statistical trends that might predict lower likelihood of response included 2+ prior therapies, R132H mutation, prior stem cell transplant, and poor risk cytogenetics. As mentioned above, the trial’s primary efficacy cohort included 125 R/R patients, but NCT02074839 also tested the feasibility of using ivosidenib in a group of patients with newly diagnosed AML who were not considered to be candidates for traditional intensive therapies. This 34-patient cohort (average age = 76.5 years) received 500 mg of ivosidenib daily and was composed largely of secondary AML cases (76%) with about half (46%) having received prior treatment with hypomethylating agents. Despite these limitations, the median overall survival for the cohort was 12.6 months. CR or CR with incomplete hematologic recovery was reached in 42.4% of patients, with 77.8% of patients remaining in remission at 1-year of follow up. Among 21 transfusion-dependent patients at baseline, 9 became transfusion independent. However, as mentioned above, IDH inhibitor therapy is not curative and does not produce permanent remissions. By a median follow up time of 23.5 months, 79% of participants had discontinued use of the drug, largely due to disease progression or adverse effects [60].

In addition to financial toxicity mentioned above, *IDH1/2* inhibitors have been characterized for their distinctive risk of differentiation syndrome. The entity, first described in the context of acute promyelocytic leukemia as leukemic cells are promoted to mature beyond states of differentiation arrest, is a life-threatening complication characterized by fever, dyspnea, hypotension, weight gain, and pulmonary infiltrates [61]. Analysis of the pivotal studies for ivosidenib and enasidenib submitted to the FDA revealed a 19% prevalence of differentiation syndrome for both agents, with increased risk associated with bone marrow blasts above 48% and peripheral blasts above 25% and 15% for ivosidenib and enasidenib, respectively [62]. Despite the serious clinical consequences of differentiation syndrome, most patients do well with supportive care. Hence, awareness and early suspicion on the part of clinicians is paramount [63]. Finally, it should be noted that neither IDH inhibitor has been approved for use by the European Medicines Agency at the time of this publication.

### 4.3. BCL-2 Inhibition

We end our discussion of recently approved therapies by highlighting venetoclax. It should be pointed out that unlike our discussion of FLT3 or *IDH1/2* above, use of venetoclax does not depend on the mutational status of its cellular target. Nevertheless, of the novel agents, venetoclax has arguably generated the most excitement and has been widely adopted by the hematology community. Venetoclax is an oral, selective inhibitor of BCL-2. As a member of several proteins that inhibit the apoptotic response, active BCL-2 binds and sequesters proteins which mediate controlled cell death. Mechanistically, venetoclax adheres to a BH3-binding groove of the BCL-2 protein, thus displacing other BH3-only binding proteins. These proteins are typically sequestered by BCL-2 and therefore their neutralization leads to a failure of apoptotic pathway activation. In the presence of venetoclax, these BH3-only proteins are available to activate Bak and Bax-mediated pro-apoptotic signaling [64,65]. First developed for use in chronic lymphocytic leukemia, venetoclax is useful in AML due to its modest selectivity for leukemia stem cells and its ability to help overcome inherent mechanisms of chemotherapeutic resistance [66,67].

Early clinical investigation into venetoclax in AML assessed the safety and clinical correlates of response in a phase II, single-arm trial of venetoclax monotherapy in 32 patients with R/R AML unfit for intensive salvage treatments. Of the 32 patients, 26 received at least one month of oral therapy. Response rate was noted to be 19%, and an additional 19% of patients demonstrated antileukemic activity that did not meet International Working Group criteria (partial marrow responses and incomplete count recovery) [68]. These initial studies garnered excitement in that venetoclax, when combined with other agents, might provide new treatment avenues for patients in whom intensive options were not feasible. The first evidence underscoring the synergistic effects of venetoclax combinations was demonstrated in treatment-naive individuals ages 60 and over, deemed ineligible for intensive therapy. Data from the expansion cohort demonstrated that 400 mg of venetoclax in combination with azacitadine or decitabine led to a CR or CRi of 70 and 74%, respectively, and an overall survival of 14.9 and 16.2 months, respectively. These data were received with excitement. Respectively, in the azacitadine and decitabine groups, the average age of patients was 72 and 75, 25% and 29% had secondary AML, and 39% and 48% harbored adverse risk cytogenetics. [20] Additional evidence came from a parallel Phase Ib/II study by Wei and colleagues that paired venetoclax with low-dose cytarabine, again in a patient population of older, newly diagnosed individuals not candidates for intensive induction therapy because of age or medical comorbidities, and yielded similar results. [21] Based on these clinical trials, the FDA approved venetoclax in combination with azacitidaine, decitabine, or low-dose cytarabine for use in AML patients who are ≥75 years of age or whose medical comorbidities exclude their candidacy for intensive induction therapy [15].

While the impressive results from seminal venetoclax trials have proven practice-changing, reported real-world experiences with venetoclax have depicted a more modest enthusiasm. In comparing a cohort of 33 patients treated off-label prior to venetoclax approval with a 33-patient cohort formally enrolled in a trial at the same institution, investigators at the University of Colorado described a 63% rate of CR/CRi for off-trial patients while CR/CRi was attained in 84.9% of those participating in the trial. This translated to a median overall survival of 381 days for those off-trial and 880 days for those on-trial. Venetoclax toxicities were observed among both groups and include cytopenias, neutropenic fever, and less commonly, pneumonia.

Despite the rapid and widespread adoption of venetoclax into clinical practice, several concerns are warranted. Issues relating to the selection of participants based on a subjective assessment of fitness are discussed above. Additionally, the FDA granted provisional accelerated approval to venetoclax combinations despite a lack of randomized clinical trial data. While intentions to reevaluate pending the release of phase III results seemed reasonable and were expected to validate fast-track approvals, phase III data for the azacitadine + LDAC combination failed to reach statistical significance for overall survival (7.2 vs. 4.1 months) [69]. This practice is problematic, since marked discordance has been observed in meta-analyses of non-randomized studies with subsequent randomized follow-up trials [70].

## 5. Emerging Therapies

### 5.1. MCL-1 Inhibition

MCL-1 is another anti-apoptotic, mitochondrial protein that has been shown to promote cellular survival across several hematologic malignancies, including non-Hodgkin lymphoma, multiple myeloma and AML. As a member of the BCL-2 family proteins, MCL-1 also works to inhibit the activation of BAX and BAK, which are key mediators of apoptosis initiation and downstream caspase activation. Counterregulation of BLC-2 family is mediated by a collection of pro-apoptotic BCL-2 homology 3 (BH3) proteins. Venetoclax is characterized as a BH3 mimetic. Recent work by the Australian group has investigated combinatorial approaches with BH3 mimetic therapy to target both BCL-2 and MCL-1. Previous pre-clinical studies have found that venetoclax resistance is in part driven by the enhanced expression of *MCL-1* and rational efforts focused on dual inhibition of BCL-2 and MCL-1 are a topic of promising investigation [71]. Namely, Moujalled and colleagues from the Australian group have described potent pre-clinical, anti-leukemic activity of a S55746, a novel inhibitor of BCL-2, in combination with S63845, an MCL-1 inhibitor. In AML mouse models and in patient-derived xenografts, synergistic, pro-apoptotic activity was apparent with this combinatorial approach. Importantly, dual-targeting of BCL-2 and MCL-1 was more selective to AML populations and generally spared healthy hematopoietic stem cells [72]. On the basis of these results, first-in-human studies with a clinical-grade MCL-1 inhibitor are now being tested in combination with venetoclax in de novo, secondary, therapy-related, or R/R AML (NCT03672695), and in combination with decitabine in patients with MDS (NCT03593915).

### 5.2. MDM2 Inhibition

The inactivation of the master tumor suppressor protein p53 is a common mechanism in the propagation of multiple tumor types [73]. This is commonly carried out by loss of function mutations in p53 or through degradation. One of the principal p53 interacting proteins, MDM2, is responsible for the E3 ubiquitin ligation and subsequent degradation of p53. Accordingly, it has been postulated that MDM2 downregulation or inactivation might stabilize wild-type p53 protein and allow the cell to make full benefit of its downstream tumor suppressor or pro-apoptotic effector functions. Idasanutlin is a selective, small molecule inhibitor of MDM2 which has since entered early phase clinical trials for R/R AML in combination with cytarabine (NCT01773408). Patients were eligible for participation regardless of *TP53* mutation status. Initial results from a phase 1b study revealed CR/CRi rates of 29% and follow up translational laboratory studies revealed that clinical response correlated to pre-treatment expression of MDM2 in blast populations [74]. Currently underway at multiple centers throughout Australia, Europe and the United States is the MIRROS trial (NCT02545283), a phase 3, double-blind, randomized 2:1 study of idasanutlin + cytarabine vs. placebo + cytarabine in patients with relapsed or refractory AML [75]. The primary endpoint for the study is overall survival in a TP53 wild-type population. Secondary endpoints will analyze CR/CRi, even-free and leukemia-free survival, ability to proceed to allogeneic stem cell transplant following therapeutic response, and will incorporate relevant laboratory correlates including measurable residual disease in post-treatment bone marrow samples and biomarkers of MDM2 expression in blast cells. The trial, which has now enrolled 447 patients from 80 institutions in 19 countries, utilizes a versatile, interim futility analysis based on a threshold of at least doubling the proportion of responders between the two treatment arms. The futility analysis was recently conducted and involved 120 *TP53* wild-type patients. MIRROS successfully met the study continuation criteria and is expected to be completed by January 2022 [76].

## 6. Conclusions

In this review, we detailed the complex molecular landscape of AML, drew attention to recent advancements in the AML therapeutic apparatus, highlighted the successes and potential shortcomings of landmark trials that led to the FDA approval of several new agents, and previewed exciting therapies that hold promise for the future. It is critical to underscore the importance of properly designed, randomized, controlled trials to appraise emerging therapies. Such endeavors, though costly and time consuming, are of greatest service to AML patients. It is incumbent upon us to bring forward therapies that can withstand the rigors of well-executed clinical investigation and not those that are retrospectively found to be of marginal benefit. Translational studies, such as the BEAT AML trial, will continue to inform rational drug development in the coming era of precision medicine. Although not discussed extensively in this review, the AML treatment landscape also presents exciting opportunities within the immunotherapy space. Novel molecules such as flotetuzumab, a CD123 × CD3 bispecific dual-affinity retargeting antibody, have shown promise in the relapsed/refractory AML disease setting [77]. Interestingly, marrow immune response has been shown to correlate with *TP53* mutations [78]. Additionally, chimeric antigen receptor T cell (CAR-T) therapies have been the subject of active investigation for several years and are currently beginning to enroll in the Phase I setting in the United States [79].

Additionally, trials such as the Australian INTERCEPT study, which adaptively move patients with disease progression after an initial targeted therapy to subsequent targeted agents based on longitudinal NGS evaluation and identification of disease vulnerabilities, are ongoing [80]. Ultimately, we feel that single agent targeted inhibitors will be of limited benefit given the heterogeneous clonal architecture of AML and the emergence of escape subclones under therapeutic selection. Future work assessing combinatorial approaches of multiple inhibitors introduced at strategic clinical time points or potent therapies with specificity for leukemia stem cell populations remain areas of unmet clinical need.

## Figures and Tables

**Figure 1 cancers-13-04646-f001:**
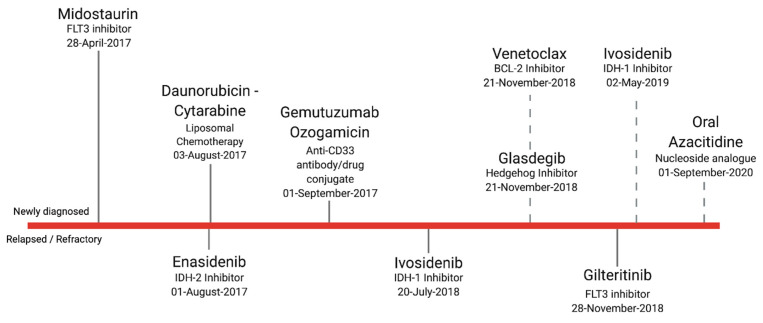
Timeline of recent U.S. FDA drug approvals for acute myeloid leukemia. Agents listed above the red bar indicate agents approved for use in the newly diagnosed setting. Those below the bar indicate approval for patients with relapsed/refractory disease. Agents with a dashed line were approved for patients felt to be ineligible for intensive treatment strategies. Of note, gemtuzumab ozogamicin is traditionally used as an adjunct to standard induction therapy but may be used as a single agent. It also carries approval for the relapsed/refractory setting but is almost always employed as part of induction for de novo disease. Lastly, oral azacytidine, the most recent addition to AML treatment options, is indicated for maintenance therapy.

**Figure 2 cancers-13-04646-f002:**
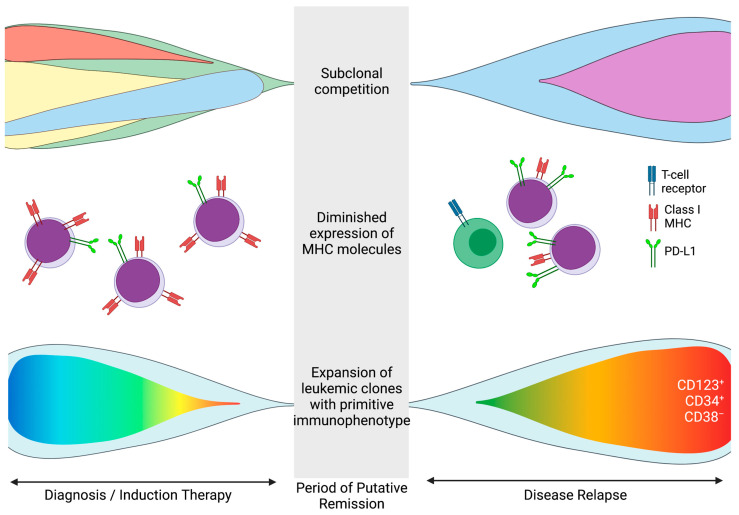
Mechanisms of acute myeloid leukemia relapse. The various schema depict three mechanisms of AML relapse delineated by NGS and flow cytometric techniques discussed in the text and include: (**top panel**) Mutational clonal heterogeneity is frequent at initial presentation. Chemotherapy-resistant subclones may persist at the time of putative remission and expand to become a dominant population at relapse. (**middle panel**) Downregulation of crucial MHC genes/molecules on leukemic cells (red receptors) or upregulation of checkpoint molecules (green receptors) during the course of treatment or at the time of relapse may result in more effective AML immune evasion and contribute to loss of graft-versus-leukemia recognition by donor T cells (green cells with blue T cell receptors). (**bottom panel**) Immunophenotyping or different-from-normal flow cytometric techniques identify leukemic populations enriched for leukemia stem cells at the time of diagnosis. At the time of relapse, expansion of clones characterized by more primitive immunophenotypic signatures (CD123^+^CD34^+^CD38^−^) may reflect the development of chemotherapy resistance.

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
