# Peer review of "Targeted Therapies for the Evolving Molecular Landscape of Acute Myeloid Leukemia"

_cancers, 2021, doi:10.3390/cancers13184646_

Round 1

Reviewer 1 Report

The revision improves the manuscript as requested.

Reviewer 2 Report

The authors made adjustments according to my requirements. 

Reviewer 3 Report

All comments have been well addressed. I have no further comments.

This manuscript is a resubmission of an earlier submission. The following is a list of the peer review reports and author responses from that submission.

Round 1

Reviewer 1 Report

I reviewed a very interesting and well-composed paper and have no major  issues with the report except one refers to figure 2, main figure of the paper:
1. Figure 2 described in the text (p.5) with separation to Figure 2A, 2B, and 2C corresponds to “top”, “middle” and “bottom” panels in the Figure 2 legend on the next page (p.6). This should be unified.  
The figure itself needs improvement. 
(i) To the upper/A panel I suggest naming "clones", eg, orange – clone A, green – clone B, etc. 
(ii) It is also unclear how “blue” clone appears in relapse if it vanished before the endpoint of remission induction; compare this with figures from quoted Jacoby et al. JCI Insight. 2018;3(5):e98962. Published 2018 Mar 8. doi:10.1172/jci.insight.98962 (ref 35); 
(iii) The middle/B panel: unclear, even after reading of comments in the text and legend, (a) what cells are presented leukemic or malignant or both, especially on relapse/right side: different shape and color. (b) Also unclear which molecules presented on the surface, probably “dark red” is MHC II while “dark cyan” is MHC I (by analogy with cited paper Christopher et al. Immune Escape of Relapsed AML Cells after Allogeneic Transplantation. N Engl J Med. 2018;379(24):2330-41 (ref 29): needs definitions in the legend.
(iv) bottom/C panel is unclear as well: what is the meaning of fields and multiple colors? the impression that AML stem cells not increased in frequency but appears only while full-blown relapse; this panel also should be corrected.
(v) Finally, what is the gray field in the middle? Is it a box for the text or a part of the timeline? Which events happen during this period? Remission? Transplantation? Or both? 
Figure 2 is an important central illustration and it should stay in the paper after suggested corrections.

"Minor" points:
On p.2 Karnofsky Performance Status presented as criteria while a few lines down used probably ECOG Performance scale (three times)(from refs 19, 20).

Author Response

1. Mention of figure 2 in the legend and the main text have been modified to refer to the "top""middle" and "bottom" panel.

i and ii. The clonal composition of the top panel has been modified, specifically with respect to the cyan/blue subclone to show relative stability/moderate expansion throughout induction with subsequent expansion at relapse. We feel that this makes the depiction at relapse to be less confusing. 

iii. The middle panel has been augmented with labels in the legend and a key in the figure to demarcate the downregulation of MHC molecules on leukemic cells at relapse and the upregulation of checkpoint molecules, which we feel helps better depict the concept of AML immune escape at relapse. The legend has been expanded upon to better explain this concept. 

iv. The legend pertaining to the bottom panel has been improved upon, explaining the depiction of differential immunophenotype at relase vs. first presentation, and the outgrowth of populations enriched for leukemia stem cells. 

v. The grey field in the middle has been labeled "Period of Putative Remission" for clarity. 

Minor points- the discrepancy between KPS and ECOG has been rectified. We appreciate the review's catching this oversight. 

Reviewer 2 Report

This is a very comprehensive review on the most promising targeted drugs currently used/under experimentation in adult AML. I have some suggestions trying to improve this very nice review.

Minor comments:

1.Since AML is also a life threatening disease in children i suggest authors to discuss same targeted drugs process development in children with AML. I think it could be very relevant trying to give an idea if same drugs might be used also in adolescents.

  1. Thanks to NGS and single cell sequencing we are retrieving novel gene/mutations involved in AML. I think this review might end with a prevision of the novel targeted therapies being expected to enter and improve AML drug-portfolio.
  2. Due to clonal evolution and to the co-existence of mutations, authors might discuss also the possibility or not to use more than one targeted therapy together, are they supposed to be synergistic? Are there any experimentation about combined treatments? Please expand this argument.
  3. Toxicity is the main issue regarding targeted therapies, please provide evidences or discuss this important issue when possible.
  4. In the conclusion authors might discuss the use of targeted therapy compared to cell-based immunotherapy: CAR-t cells or monoclonal antibody can replace/improve/ be better than targeted drugs before HSCT for example? Or in R/R AML? Discuss this important issue.

Author Response

  1. We agree with Reviewer 2’s point that FLT3 mutant AML is also highly relevant in pediatric and young adult populations and have added a paragraph at the end of the FLT3 section providing a brief overview of ongoing exploration.
  2. We share the excitement with the review of novel treatment in the development pipeline whose design has been largely inform by precision sequencing strategies. We call attention to two paragraphs in at the end of the paper - one focused on emerging MCL-1 inhibitors, and the other focused on MDM2 inhibitors – that are currently in development. This work, led largely by Dr. Konopleva at MD Anderson Cancer Center in Houston is currently accruing for multicenter enrollment.
  3. This point regarding the emergence of clonal resistance is appreciated and noted, specifically with regards to the development of clinical protocols testing sequential use of targeted therapies. To this end, our conclusion has been augmented with description of the INTERCEPT Study, an Australian study using sequential, NGS-guided targeted therapy strategies.
  4. This point is well-taken and appreciated. To improve discussions of drug toxicity for the pertinent agents described, a paragraph discussing differentiation syndrome has been added to the IDH1/2 section. Additionally, a paragraph on the real-world experience with venetoclax, including mention of common toxicities, has been added as the penultimate paragraph to the venetoclax section.
  5. As we anticipate immunotherapeutic strategies for AML will likely be the topic of another article in this special series, we did not address it here but have added brief mention of two approaches, CAR-T and bispecific antibody therapy with flotetuzumab, to the conclusion.

Reviewer 3 Report

This is a review paper focused on the treatment of AML, especially in patients who are not suitable for conventional induction therapy or bone marrow transplantation in terms of age or medical condition. Such orientation is important because there are several new approved therapeutic protocols, especially for these cases. The informational value of the article cannot be denied, but the authors did not use all the possibilities that this topic gives them.
Therefore, I suggest that they supplement the work according to the following comments:
1. Please briefly describe the spectrum of genotypic and phenotypic characteristics used to diagnose AML subtypes (possibly using a scheme or table).
2. Please compare AML with other blood malignancies arising from improper myeloid differentiation.
3. Please devote more space to describing the mechanisms of action of individual newly approved drugs on AML together with their undesirable side effects.
4. Please pay attention to the possibility of developing cell resistance to individual proposed drugs, possibly by creating a separate chapter, which would discuss the various mechanisms of multidrug resistance already known today.
Minor point.
1. Please write references to the text in the format "XXX [number]." instead of "XXX. (number)".
2. Figure 2 does not have parts A, B and C to which the text on page 5 refers.

Author Response

1 & 2:  We have added citations to our discussion of the genotypic and phenotypic properties of AML in the section “Molecular landscapes of de novo vs. R/R AML,” specifically with work by Papaemmanuil depicting a novel genomic classification novel for disease. Due to word limitations and having been tasked by the editors of this special series to focus on new drug approvals for AML, we unfortunately must forego head to head comparisons of AML to other myeloid malignancies in order to devote proper attention to nuanced discussion of recent therapeutic approvals.

  1. We appreciate this feedback and agree that the article would be improved by addition of details regarding mechanisms of the drugs discussed. Several sentences have been added to our discussion of the targeted agents we selected to highlight (venetoclax, IDH inhibitors) that supplements details on their mechanisms of action and overviews common adverse events that occur (such as differentiation syndrome with IDH inhibitors). The IDH section has been augmented to include a more detailed discussion on the R-2-HG oncometabolite that is formed in the presence of IDH inhibition and how restoration of this oncometabolite to normal levels overcomes the differentiation block that is critical to leukemogenesis in IDH-mutated patients. To the venetoclex section, we have added reference to a “real-world” experience published by Pollyea and colleagues at the Univ. Colorado that describes the observance of lower response and OS rates in their practice than were seen in the venetoclax trials, in addition to issues of cytopenias, neutropenic fevers and pneumonias.

  1. While word limits/spatial constraints prevent the possibility for a separate chapter/article on this topic, we have added to the conclusion a description of the ongoing INTERCEPT trial in Australia/New Zealand which is positioned to evaluate the resistance phenomena with targeted therapies.

Minor points:  1. References have been formatted accordingly. 2. Figure 2 has been modified to consistently include "top" "middle" "bottom" references in the text and figure legend. 

Reviewer 4 Report

The manuscript entitled “Targeted Therapies for the Evolving Molecular Landscape of Acute Myeloid Leukemia” by Ahmadmehrabi et al. deals with a topic that is interesting for clinicians and translational researchers. Precision oncology and especially targeted therapies in AML are of growing interest also because of recently approved novel drugs for treatment.

Comments:

1) Paragraph 2 “The Problem with ‘Fitness’” is a very important topic, especially, since AML is a disease of older people as mentioned by the authors. However, I would recommend including/discussing some additional models that are used to define eligibility (see also Palmieri Cancers 2020, Ferrara Leukemia 2013; Walter JCO 2011; Wheatley BJH 2009).

2) The term “Leukemia stem cell population” should not be used synonym with CD34+/CD38-. Several studies showed that LSCs can also be found in the other populations.

3) Mutations in the IDH genes and their prognostic role in AML are still an active research topic. The authors claim that “general expert consensus […] is consistent with mutations in IDH1/2 conferring intermediate risk”. I think this conclusion needs a citation that supports the statement.

4) IDH inhibitors are a novel and promising class of drugs that allow a targeted therapy. However, both drugs, Enasidenib and Ivosedenib, have not been approved by the European Medicines Agency (EMA). The author should mention this fact and describe the limitations that led to the withdrawal.

5) Figure 2 is not labeled with a, b, and c. 

Author Response

  1. We appreciate the reviewer’s comments on the importance of the section regarding the issues of determining “fitness” in AML. We found the review article by Palmieri published in Cancers and suggested by Reviewer 4 to be very comprehensive and have provided reference to the paper for readers who wish to learn more.

  1. We acknowledge the reviewer’s point regarding the distinction between AML cells that are CD34+/CD38- and the term “leukemia stem cell” and the text has been modified to describe them with adjectives as “immature leukemia cells” or “stem-cell enriched CD34+CD38- populations.

  1. The reviewer’s comments are appreciated and the text has been modified to express that IDH-mutated AML representing intermediate-risk disease is our opinion, with reference provided.

  1. In the past paragraph of the IDH section, it has been noted that to date, enasidenib and ivosidenib lack approval in Europe.